# Thermal Preprocessing of Rapidly Solidified Al 6061 Feedstock for Tunable Cold Spray Additive Manufacturing

Baillie Haddad [1], Bryer C. Sousa [2], Kyle Tsaknopoulos [2], Victor K. Champagne, Jr. [3], Richard D. Sisson, Jr. [2], Aaron Nardi [1] and Danielle L. Cote [2,4,*]

1 VRC Metal Systems, Rapid City, SD 57719, USA; baillie.haddad@vrcmetalsystems.com (B.H.); aaron.nardi@vrcmetalsystems.com (A.N.)
2 Department of Mechanical & Materials Engineering, Worcester Polytechnic Institute, Worcester, MA 01609, USA; bcsousa@wpi.edu (B.C.S.); kltsaknopoulos@wpi.edu (K.T.); sisson@wpi.edu (R.D.S.J.)
3 U.S. Army Research Laboratory, Aberdeen Proving Ground, Adelphi, MD 21005, USA; victor.k.champagne.civ@mail.mil
4 Chemical Engineering Department, Worcester Polytechnic Institute, Worcester, MA 01609, USA
* Correspondence: dlcote2@wpi.edu

**Abstract:** In this work, the influence of thermal pre-processing upon the microstructure and hardness of Al 6061 feedstock powder is considered through the lens of cold spray processing and additive manufacturing. Since solid-state cold spray processes refine and retain microstructural constituents following impact-driven and high-strain rate severe plastic deformation and bonding, thermal pre-processing enables application-driven tuning of the resultant consolidation achieved via microstructural and, therefore, mechanical manipulation of the feedstock prior to use. Microstructural analysis was achieved via X-ray diffraction, scanning electron microscopy, transmission electron microscopy, electron backscatter diffraction, energy dispersive spectroscopy, and differential thermal calorimetry. On the other hand, nanoindentation testing and analysis were relied upon to quantify pre-processing effects and microstructural evolution influences on the resultant hardness as a function of time at 540 °C. In the case of the as-atomized powder, $\beta$-$Mg_2Si$-, Al-Fe-, and Mg-Si-type phases were observed along polycrystalline grain boundaries. Furthermore, after a 60 min hold time at 540 °C, Al-Fe-Si-Cr-Mn- and Mg-Si-type intermetallic phases were also observed along grain boundaries. Furthermore, the as-atomized hardness at 250 nm of indentation depth was 1.26 GPa and continuously decreased as a function of hold time until reaching 0.88 GPa after 240 min at 540 °C. Finally, contextualization of the observations with tuning cold spray additive manufacturing part performance via powder pre-processing is presented for through-process and application-minded design.

**Keywords:** cold spray; rapid solidification; heat treatments; gas-atomization; microscopy; nanoindentation; powder

## 1. Introduction

Traditional metal additive manufacturing techniques, such as selective laser sintering, electron beam melting, and laser engineered net shaping, notably depend on powder-centric variables such as the particle size distribution, morphology, and flowability of powders used in said processes. As reported previously, the powders' sphericity and morphology—which captures two powder-centric variables for traditional powder-based metal additive manufacturing methods—will affect the powder's flow through the powder feeder in such systems [1]. These variables matter in laser-engineered net shaping because a specific size range of powder, 36–150 μm in diameter, must be continuously injected onto the powder bed in a controlled fashion as the laser passes over it [2]. In contrast, the size distribution and morphology matter to a lesser degree (although they remain significant and non-negligible, as discussed by Valente et al. in [3] and pointed out by Hussain et al.

in [4]) for the cold spray additive manufacturing (CSAM) process. Nevertheless, in CSAM processing, small particles may not bond to the respective target substrate under a given set of processing conditions for one or both of the following reasons: (i) fine particles can suffer from an inability to sufficiently plastically deform (due to increased yield strengths being associated with smaller atomized particles, as they experience higher solidification rates) and therefore bond with the substrate; and (ii) from the bow shock effect [5,6].

One should note that successful attempts at utilizing sub-10 μm diameter feedstock particles have been reported in the literature [7]. When particles are too large, they will not reach the critical impact velocity due to their relatively increased mass on a per-particle basis [8]. However, for those particles belonging to the appropriate range of feedstock powder particulate sizes, one of the essential powders for CSAM is the microstructure (including microstructural crystallinity and intermetallic constituents and phases). The microstructure matters so much for CSAM because CSAM is an entirely solid-state process; therefore, the initial properties and phases within the microstructure of an alloyed metallic powder greatly influence the final properties of the CSAM consolidated material. Most of the literature suggests that the phases present in the powder remain in the CSAM material.

To date, gas-atomized aluminum powder undergoes thermal pre-processing to degas the powder prior to consumption during manufacturing or modify the as-atomized internal microstructures. While powder degassing remains a common industrial practice prior to introducing the feedstock into powder-based manufacturing systems of relevance [9,10], thermal pre-processing for microstructural modification of gas-atomized powder has historically been of limited interest to the materials processing sector because conventional powder metallurgy methods remain, at least in large part, dependent on complete or partial melting; thus, generally nullifying the need for understanding the as-atomized microstructure, since particulate melting remains intrinsic to the respective methods themselves [11]. That said, thermal processing of heat-treatable alloyed aluminum systems, including Al 6061, remains sensitive to the solutionizing temperature, heating rate, quench rate, aging temperature (if precipitation hardening is sought after), and hold time. At the same time, wrought- or bulk-inspired heat-treating parameters are non-transferrable to rapidly solidified, highly non-equilibrium, and gas-atomized alloyed aluminum particulates [12]. Such non-direct transference follows from the fact that the size of the particles and the resultant solidification structure enable much more rapid atomic diffusion along sub-granular and granular boundaries, as detailed in [13]. As a result, one must formulate and identify appropriate thermal pre-processing parameters and their significant influences on the internal microstructures of gas-atomized Al 6061 to properly understand and approach the implementation of thermal pre-processing of Al 6061 feedstock for optimized and tunable CSAM processing and consolidated material performance. In turn, such matters were considered herein.

The aim of heat treating particulate CSAM feedstock significantly deviates from those traditionally affiliated with classical thermal or thermomechanical processing. More specifically, thermal pre-processing of CSAM feedstock stems from the desire to tune the resultant consolidated components' performance and enhance the particulate deformability and, therefore, suitability for CSAM [14]. Increasing the deformability of the feedstock microparticles lowers the critical impact velocity required for particle–substrate bonding. At the same time, one may consider that the previous work has highlighted that powder particles of variable diameters for a single batch of bulk powder may very well maintain microstructural intermetallic constituents that are notably variable as a function of microparticle diameter [15]. Hence, heat-treating or thermally pre-processing feedstock for CSAM can also enable greater quality assurance and normalization of the internal phase compositions, concentrations, and the like, across the range of particle sizes able to be processed via CSAM, especially when compared with the aforenoted variability of the as-atomized condition.

When applying heat treatments, two main microstructural effects are those related to grain size and the phases present. Firstly, holding an alloy at elevated temperatures for a

considerable time will cause grain coarsening. Grain growth is an important consideration when heat treating any metal because of the Hall–Petch relationship, where the strength of the alloy is inversely proportional to the square root of the grain size. The second consideration when heat treating aluminum alloys centers upon the phase transformations at elevated temperatures. Two strengthening mechanisms in aluminum alloys are solid solution strengthening and precipitation hardening. During a solutionization heat treatment step, alloying elements will partially or wholly dissolve into the aluminum matrix, causing a supersaturated solid solution (SSS) to form. This causes local distortion in the aluminum lattice and will thus increase the material's strength by obstructing dislocation motion. An appropriate solutionization temperature is 5 °C below the solidus temperature. The alloy is then quenched to room temperature to preserve the high-temperature composition of the SSS [16]. Unfortunately, it is difficult to simply apply heat treatment schedules from literature to the powder for cold spray, as it has been shown that powders will not respond to heat treatments in the same fashion as rods, sheets, plates, for example [12,17]. Therefore, powders have no universally accepted and specific heat treatment schedule to date.

It has not yet been determined which thermally processed condition and processing parameters are the most suitable for gas-atomized Al 6061 feedstock-based CSAM. Accordingly, the present work presents experimentally observed thermal pre-processing effects that will enable continued research and development throughout the CSAM community in identifying a thermal processing window for alloyed Al 6061 powders. Furthermore, the present work focused on homogenizing or solutionization of the gas-atomized Al 6061 powder microstructures, as there has been an initial success in cold spraying the powder in this condition. An as-atomized powder can be considered a cast structure where the rapid cooling or solidification rate causes intermetallic secondary phases to nucleate along interdendritic regions or grain boundaries. However, a homogenous solid solution is a more desirable microstructure for many applications. Sheppard and Raghunathan conducted several different homogenization treatments on Al 5056 samples to study the effects on the microstructure [18]. The homogenization treatment aims to put the segregated solute atoms into the solution. It is hypothesized that homogenization times for the powder will be less than what is seen in the literature because the grain sizes are much smaller than the cast materials studied. Therefore, there is less distance for the atoms to diffuse.

As discussed in the aforementioned introductory remarks, the microstructural and mechanical properties of non-equilibrium, rapidly solidified, and gas-atomized aluminum alloys are of interest to the CSAM materials consolidation research and engineering communities [19]. As previously mentioned, such interest arises from the fact that CSAM processing is microstructurally retentive (although refined), and the mechanical properties of particulate feedstock directly affect the critical parameter known as the critical impact velocity required to achieve mechanical interlocking as well as intimate metallurgical bonding between the powder particles and a target substrate material together with the particle–particle bonding too [20]. Given the well-established fact that the mechanical properties of a metallic material are directly dependent upon the metal's microstructure and processing history [21], one can pre-process polycrystalline, rapidly solidified, CSAM feedstock powders prior to use in the CSAM process to finely-tune and control the resultant consolidated materials performance, behavior, and properties [22].

Rapidly solidified, gas-atomized alloyed aluminum (and other matrix materials) powders can be preprocessed before consumption during the CSAM materials consolidation process. Preprocessing methods include milling [23], vacuum degassing [9], thermally assisted degassing [24], thermal heat treating [25], plasma spheroidized [26], granulation [27], spray drying [28], plating [29], and more [30]. Degassing, milling, and thermal heat-treating capture most preprocessing methods that have been researched and developed in the CSAM literature. However, degassing, milling, and thermal heat treatments are not universally applicable across gas-atomized aluminum alloys. For example, degassing preprocessing parameters must be optimized according to a trial-and-error approach, even when two powders are of the same nominal composition [31]. At the same time, thermal preprocess-

ing requires knowledge of the powders' composition and classification to target desired secondary phase formation and deleterious phase dissolution or achieve a homogenized and solutionized state [32].

Growing interest exists surrounding the industrial implementation of thermal pre-processing of CSAM particulate feedstock materials [33]. Therefore, this work considers how experimental characterization, coupled with a previously developed through-process framework for aluminum alloy CSAM [34], can be invoked to control CSAM particulate feedstock mechanical properties and microstructures for improved CSAM consolidations performance in service and optimal processing parameter selection. Beyond influencing the critical velocity required for bonding, among other noteworthy CSAM parameters, thermal pre-processing of CSAM alloyed aluminum feedstock systems can be utilized to address CSAM component ductility and toughness without further post-processing. Thus, one way to improve ductility is to thermally pre-process powder before CSAM deposition. This is carried out for two reasons: hydroxides are removed from the surface of the powder, which is a typical degassing step for powder processing, and secondly, the microstructure is modified from the as-atomized condition to a more "sprayable" condition.

## 2. Materials and Methods

The gas-atomized Al 6061 powder utilized during this research was procured from F. J. Bromann & Co. (Harvey, LA, USA). By direct current plasma emission spectroscopy-based analysis, performed per ASTM E 1097-12 and outsourced to Luvak Laboratories, Inc. (Boylston, MA, USA), the chemical composition of the gas-atomized Al 6061 powder was unveiled. More specifically, the measured chemical composition of the gas-atomized Al 6061 CSAM feedstock was as follows: 0.11 wt.% Cr, 0.26 wt.% Cu, 0.28 wt.% Fe, 0.922 wt.% Mg, 0.078 wt.% Mn, 0.024 wt.% Ti, 0.02 wt.% Zn, 0.591 wt.% Si, and 97.7 wt.% Al. Said chemical composition falls within the standard Al 6061 compositional tolerances and ranges specified for each alloying element [16].

Rapidly solidified, gas-atomized aluminum alloy powder particle properties have previously been shown to depend on particle diameters [35]. Accordingly, the powder was sieved before characterization was performed. Concerning the powder sieving, fine particles of less than 20 μm in diameter were removed by introducing a stream of nitrogen gas through a modified fluidized bed wherein the fine particles were ejected from the bulk batch of powder and the fluidized bed's gas stream. The removal of said fine particulates ensured that the sieve meshes employed would not clog and also ensured that amorphous powder particles were removed from the analysis in light of prior reports that made mention of amorphous particulates forming from the very rapid cooling rates associated with sub-10 μm sized microparticles manufactured via gas atomization [36]. Using a mechanical sieve, which was equipped with interwoven stainless-steel wire-based meshes, resulted in sieve size categories ranging from 25–32 μm, 32–38 μm, 38–45 μm, 45–53 μm, and 53–63 μm in particulate diameter. Particle size distributions of the sieved gas-atomized Al 6061 powder—that is, particle size distributions for each of the size categories—were determined by way of laser diffraction-based analysis at the United Technologies Research Center (now part of Raytheon Technologies Corporation and located in East Hartford, CT, USA), as shown in Figure 1.

For studying the heat-treated powders, small samples of powder (approximately 30 mg) were thermally processed in a TA Instruments, Inc., (part of Waters Corp. and located in New Castle, DE, USA) Q20 differential scanning calorimeter (DSC). As needed, heat treatment times, temperatures, and heating rates are stated hereafter for each respective sample in the Results and Discussion sections. Still, in DSC analysis, the DSC data were collected with a heating rate of 5 °C min$^{-1}$ and scanned from 20–530 °C. The DSC testing atmosphere comprised nitrogen with a flow rate of 50 mL min$^{-1}$. A Tzero Al pan and lid were used, with an empty Tzero Al pan and lid for the reference sample. While the generally favorable and hermetically sealable Tzero Al lid option was not used herein, as has been performed in similar studies, to ensure that degassed oxygen-rich and hydroxide-

rich volatilized outgassing species could potentially escape the pan housing the powder, the hypothetical risk of nitriding the particle surfaces was shown to have been avoided during heat treating via electron microscopy and localized EDS-based elemental analysis of the DSC-processed powders.

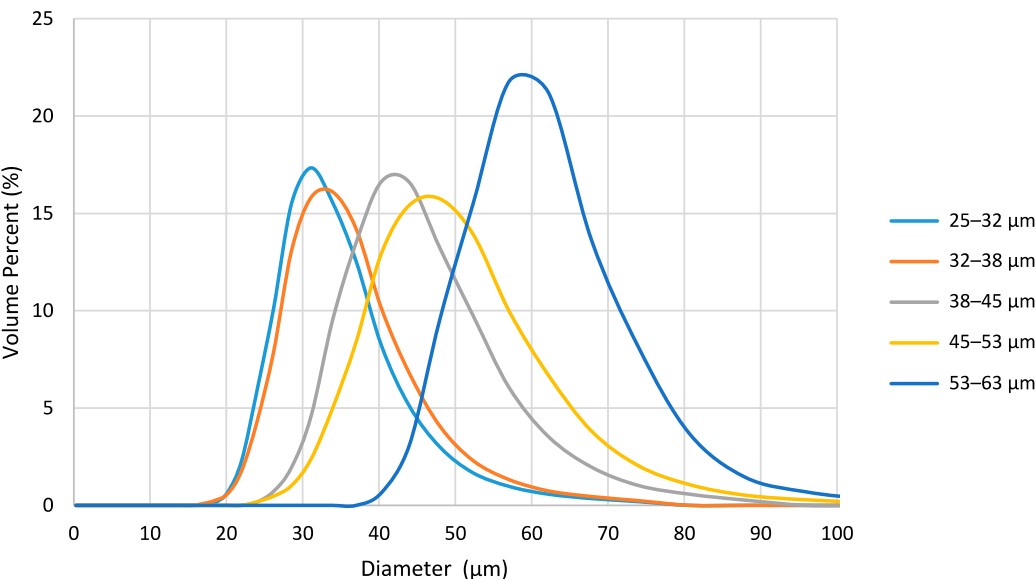

**Figure 1.** Particle size distribution of the sieved, gas-atomized Al 6061 powder was conducted by laser diffraction analysis.

Nevertheless, the powder was removed from the DSC immediately after the allotted time and rapid quenching in the DSC sample reservoir. Feedstock powders were cold mounted in epoxy, ground with 800-grit silicon carbide paper, and then mechanically polished using a Struers (part of Roper Technologies and located in Ballerup, Denmark) Tegramin-20 auto-polisher to a final 0.05 μm silica polish. For microstructural examination, the gas-atomized Al 6061 powders were etched with a 0.5% hydrofluoric acid solution for 60 s.

The microstructural examination was performed with a NIKON Epiphot 200 light optical microscope (Nikon Corporation, Minato City, Tokyo, Japan), and a JEOL, Ltd. (Akishima, Tokyo, Japan), JSM-7000F field emission scanning electron microscopy (FESEM) with an Oxford (Oxford Instruments, Abingdon, Oxfordshire, UK) X-MAXN silicon drift detector for energy dispersive spectrometry (EDS). For further examination, a TEM lamella was cross-sectioned and prepared by a Thermo Fisher Scientific (Waltham, MA, USA) Helios NanoLab G3 UC DualBeam scanning electron microscope and focused ion beam (SEM/FIB) at the Institute of Materials Science (IMS) at the University of Connecticut (Mansfield, CT, USA). The lamellas were imaged in a Thermo Fisher Scientific FEI Talos F200X scanning electron microscope (STEM), also housed at UConn's IMS. Copper was not included in the EDS analysis because the sample was mounted on a copper grid. This would cause more copper to appear on the EDS map than in the material. The limitations imposed by using such a grid were considered as detailed hereafter in the Discussion section of the present manuscript. Transmission electron backscatter diffraction (t-EBSD) was conducted in a JEOL JSM-7000F field emission SEM at a voltage of 30 V. Phase fraction and precipitate size measurements were performed in the image analysis software Olympus Stream Essentials (Olympus Corporation, Japan). Automatic thresholds were employed. Area fraction results from image analysis were converted to volume fractions using a method described by Corti et al. in [37].

Conventionally static nanoindentation hardness measurements of the gas-atomized Al 6061 feedstock particulates, as a function of processing condition and even size, were performed using a Keysight Nano Indenter G200 (Keysight Technologies, Santa Rosa, CA,

USA). However, one ought to note that the G200 nanoindentation system has since been acquired by the KLA Corporation and rebranded as The Nano Indenter® G200 system under the umbrella of KLA Instruments (part of the KLA Corporation, Milpitas, CA, USA, and has incorporated Nanomechanics, Inc. (Oak Ridge, TN, USA), too). In any case, the Keysight model of the nanoindentation system, formally known as the Keysight Nano Indenter G200, was equipped with a Berkovich diamond pyramidal nanoindenter tip from Micro Star Technologies (Huntsville, TX, USA). Nanoindentation measurements were made at an indentation depth of 250 nm, which compliments the recommended depth limits for proper indentation testing of metallurgical powder particles mounted in compliant matrix materials by Sousa et al. [36]. For each of the processed gas-atomized Al 6061 powder conditions, at least 30 nanoindentation measurements were recorded. The displacement resolution of the Keysight Nano Indenter G200 was less than 0.01 nm, and a passive antivibration isolation mount below the G200 nanoindenter ensured well-controlled testing conditions.

## 3. Results and Discussion

One of the most critical features of powder feedstock for CSAM processing is the microstructure [38]. CSAM is an entirely solid-state process, so feedstock powder particles' initial properties and microstructural constituents dramatically influence the CSAM consolidated material [39]. Stated another way, the phases present in the powder remain in the CSAM material. This was evident by formulating Figure 2a,b. One may readily note that the XRD patterns in Figure 2c suggest that no significant phase transformations occurred during CSAM-based processing of the as-atomized Al 6061 studied during the present work. Accordingly, when SEM analysis was considered alongside the XRD analysis presented in Figure 2c, the same two phases, which were indicated by the arrows in the micrograph of the powder presented in Figure 2a, can also be found in the corresponding CSAM sample presented in Figure 2b.

As already mentioned, Al 6061 has been highly regarded and considered within the CSAM community since the 2000s and remains as such [40]. Still, consideration of the current state of the literature surrounding the use of rapidly solidified and gas-atomized Al 6061 particulate feedstock within CSAM processing unveils an ongoing discrepancy surrounding the microstructural solidification type or class such rapidly solidified alloyed aluminum microparticulate powders belong. Stated otherwise, researchers have claimed that gas-atomized metallic droplets' rapidly solidified, non-equilibrium, and undercooled nature may vary between dendritic, mixed dendritic/cellular (also known as compound), and cellular, as evidenced by the work of Behulova et al. in [41]. Considering such microstructural variability and sensitivity to processing conditions, etc., during atomization suggests that care must be taken when attempting to directly compare one reported microstructure with another for the same nominal alloy system. Nevertheless, one ought to still consider the claims reported within the academic literature of relevance.

Per the need for continued consideration of the literature reported to date surrounding gas-atomized Al 6061 powder particle microstructures, Bedard et al. was considered first herein. More to the point, Bedard et al. claimed to have observed "cell-like solidification microstructures" in their Al 6061 feedstock [42]. Beyond the work of Bedard et al., Ernst et al. purported that "cellular-dendritic", i.e., compound-like, solidification microstructures were observable in another gas-atomized Al 6061 powder [43]. However, according to Vijayan et al., gas-atomized Al 6061 powder particles reportedly maintained a pronounced cellular solidification microstructure that was identified as being more cellular than the cellular/dendritic or cell-like microstructures reported by Bedard et al. and Ernst et al. [44]. At the same time, Sabard et al. reportedly noted a dendritic microstructure during their research concerned with gas-atomized Al 6061, even though they state that a compound-like or "cellular/dendritic microstructure" follows from the high cooling rates achieved during the solidification of the particles in the introduction to [45]. Work by Evans et al. also asserted that a "cellular rapid solidification structure" could be found within gas-atomized

Al 6061 [46]. Nevertheless, contemporary work by Wei et al. focused on identifying equiaxed grains within their as-atomized Al 6061 feedstock rather than compound or cellular grains, which offers an alternative example of the rapidly solidified structures reported upon to date within the scholarly literature of relevance [47].

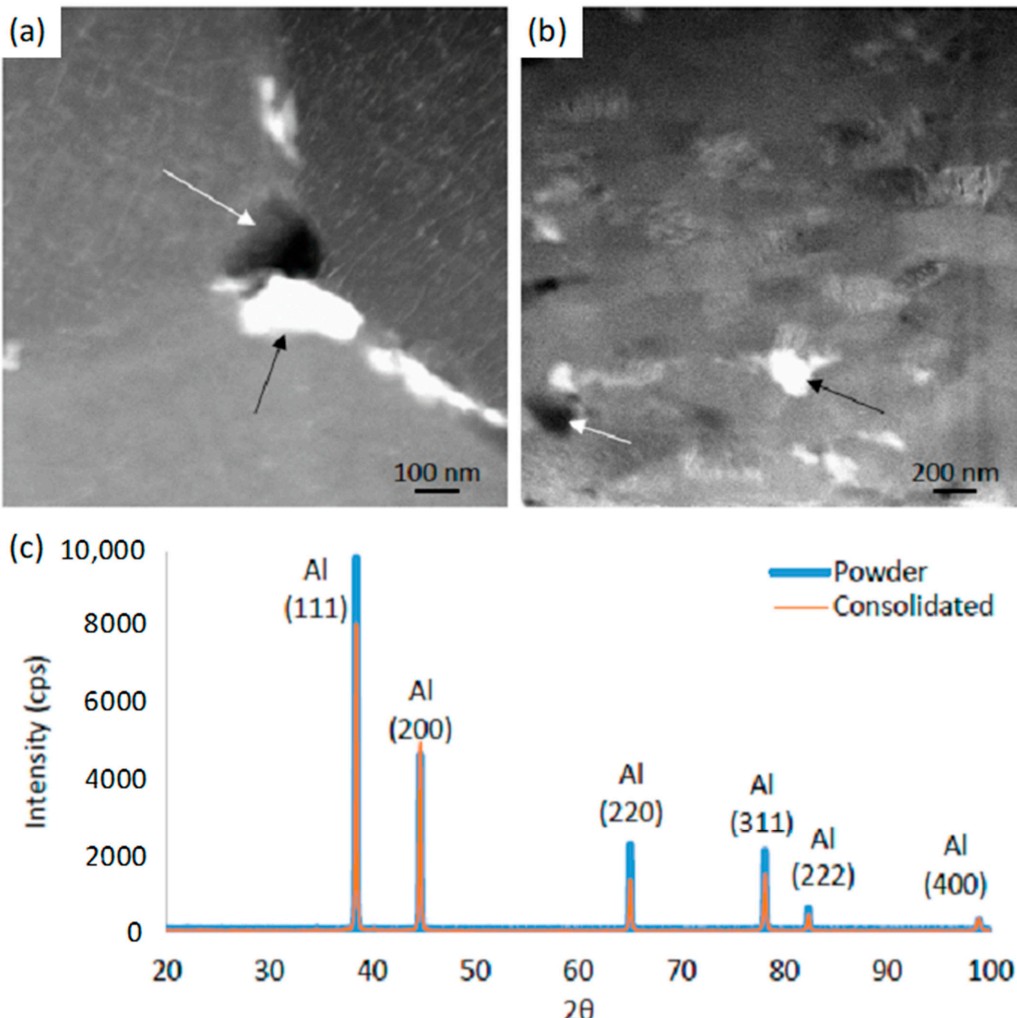

**Figure 2.** STEM images of (**a**) Al 6061 powder with a scale bar of 100 nm and (**b**) Al 6061 cold spray with a scale bar of 200 nm. The white arrows indicate the dark phase in both images, and the black arrow indicates the white phase. (**c**) presents XRD pattern for Al 6061 powder and consolidated material.

In keeping with the consideration of alternative interpretations of the resultant microstructural classification of gas-atomized Al 6061 and observed microstructural solidification types reported upon as of late, the recent work of Sousa et al. may be considered next. Interestingly, Sousa et al. demonstrated how secondary/primary dendritic solidification modeling can accurately predict the resultant grain size of gas-atomized Al 6061 particles as a function of the atomization gas utilized, molten droplet/particle diameter, cooling rate [12]. The notable agreement between the computed and measured effective grain sizes attests to dendritic or compound-based solidification of rapidly solidified gas-atomized Al 6061. One may also note that Sousa et al. not only substantiated the dendritic solidification class of gas-atomized Al 6061 in [12]; rather, Sousa et al. also substantiated the dendritic nature of the non-equilibrium microstructure associated with gas-atomized Al 6061 in [48] and [36] as well. Additional works of relevance also serve to either substantiate or detract from the suitability of a dendritic solidification frame of reference, as put forth by Sousa et al. (and Belsito in [49]), for gas-atomized Al 6061. For example, Rokni et al. identi-

fied the polycrystalline microstructure in Al 6061 particulates gas-atomized as equiaxed with low-angled sub-grains comprising high-angled grains [50]. However, as a counter example, work by Tsaknopoulos et al. and Walde et al. identified the possibility of cellular precipitation or granular nucleation in [51] and [52], respectively.

With the discussion above in mind, it stands to reason that it is still unclear from the literature's interpretations if commonly gas-atomized aluminum powder microstructures are primarily comprised of dendrites, a cellular structure, or a compound microstructural constituent. For example, dendritic solidification microstructures were also observed in a non-6061 alloyed aluminum powder [53]. Thus, for the present work, a preliminary EBSD study was conducted on the as-atomized Al 6061 powder studied herein to give more information about the grain structure in the powders. Recall that the powder particles were polished, etched, and viewed as cross-sections of the respective particulates via SEM and optical light microscopy. The Materials and Methods section noted that the mounted and polished gas-atomized Al 6061 cross-sections were etched with a 0.5% hydrofluoric acid solution. Grain size parameters were measured from SEM and optical microscopy micrographs using the ASTM E112-13 intercept method [54]. It was revealed that smaller particles contain smaller grains while larger particles contain larger grains. Such an observation may be relatively intuitive because larger particles have been shown to solidify with lower cooling rates, whereas smaller particles solidified with greater cooling rates during the gas-atomization production process [55].

For the present work, a preliminary EBSD study was conducted on the as-atomized Al 6061 powder studied herein to yield more information about the internal polycrystalline grain structure associated with the powders. As shown in the inverse pole sub-image of Figure 3, larger, high-angled grains contain smaller sub-grains, or low-angled grains, with the same orientation. Other individual grains with different orientations are not contained within larger grains. Herein, the term "grains" essentially encapsulates sub-granular low-angled grains since (i) high-angled grains were found to house multiple low-angled and similarly oriented grains, (ii) have previously been shown to act as the adequate grain size that dictates mechanical properties, and (iii) matches modeled secondary dendrite arm spacing (SDAS) as a function of cooling rate and therefore gas-atomized powder particle size [12,34,36]. Since sample preparation was considered essential for EBSD analysis, the sample associated with Figure 3 was thinned through focused ion beam milling.

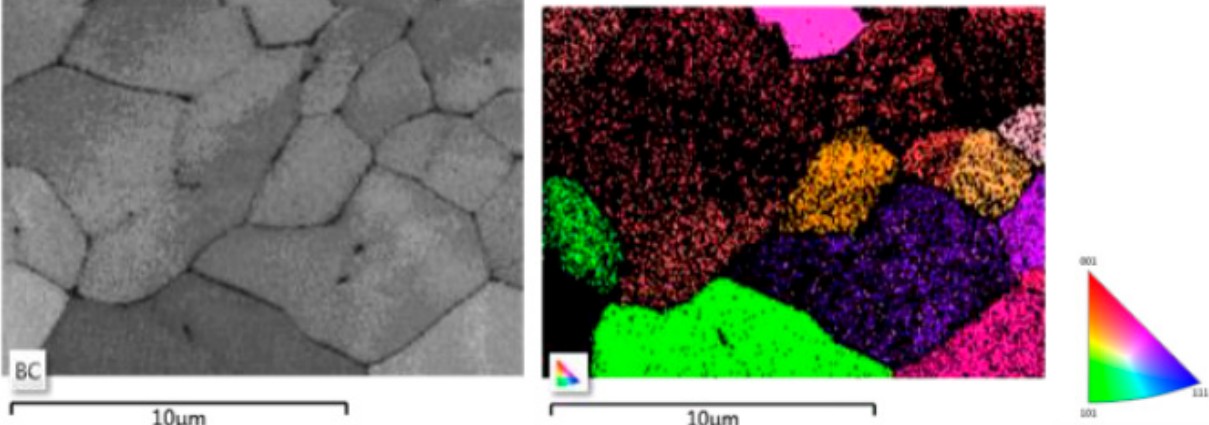

**Figure 3.** EBSD analysis of an as-atomized Al 6061 powder particle. The grayscale micrograph presents a band contrast image of the analyzed cross-sectional region considered the scanning area for EBSD-based characterization. The colored micrograph presents the inverse pole figure color map from the EBSD analysis. The Miller indices (001, 101, and 111) are also identified next to the inverse pole figure.

Beyond the isolated realm of microstructural polycrystalline solidification class identification and grain size/orientation analysis, STEM has also emerged as a beneficial and experimentally rooted characterization tool for the study of gas-atomized, rapidly solidified microparticles in general, as was highlighted by Srinivasan in [56], and Al 6061 powders in particular herein. More to the point, STEM is suitable or adequate to unveil the possible identities and approximate compositions of secondary intermetallic precipitates along the sub-grain and high-angled grain boundaries within the gas-atomized feedstock. Moreover, McNally previously demonstrated the well-suited nature of STEM to compare gas-atomized aluminum alloyed intermetallic phases with the equilibrium and non-equilibrium predictions via computational thermodynamics and computational kinetics of the secondary phases for the experimentally determined chemical composition of the powder. Stated otherwise, the respective chemical composition of the bulk powder had been utilized as an input for computational thermodynamic and kinetic predictive analysis of prospective phases and their potential volume fractions [57]. In any case, Figure 4a captures fine precipitates along the grain boundaries, around 100 nm to 200 nm in size. The white arrow within Figure 4 points to what was labeled as the "white phase". Similarly, the dotted white arrow in Figure 4 points to what was labeled as the "gray phase", and the black arrow identifies what was labeled as the "dark phase". Figure 4b presents a higher magnification micrograph of the phases along the grain boundaries affiliated with a region captured in Figure 4a.

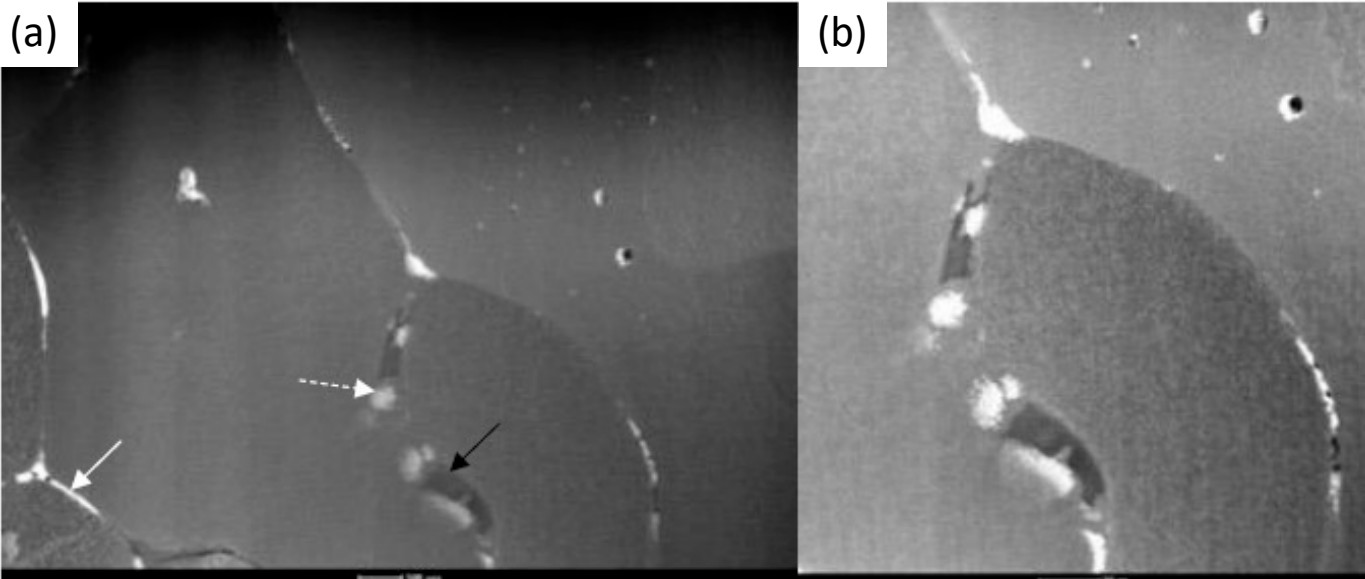

**Figure 4.** (**a**) presents a STEM micrograph of an as-atomized Al 6061 powder particle showing different precipitates, intermetallic phases, secondary phases, or dispersoids along the rapidly solidified polycrystalline sub-granular boundaries, as identified by the solid black, dotted white, and solid white arrows embedded within the micrograph. (**b**) presents a STEM micrograph of the as-atomized Al 6061 microparticle at a greater magnification such that the proximity between the sub-grain boundaries and the secondary intermetallics located along said sub-granular boundaries. Note that the scale bars affiliated with (**a**,**b**) are 500 nm. Therefore, the precipitates along the polycrystalline boundaries are 100 nm to 200 nm in size.

The average composition, in atomic percent, was assessed for each of the phases associated with Figure 4 and was provided in Table 1. The dark phase was identified as a form of $\beta$-$Mg_2Si$; however, a measured diffraction pattern did not identify the $\beta$-$Mg_2Si$ intermetallic as the stable $Mg_2Si$ phase. Said non-stable $\beta$-$Mg_2Si$ was identified as a metastable $\beta$-$Mg_2Si$ precursor precipitate phase formed during gas-atomization of the Al 6061 CSAM feedstock powder. The identification of a metastable form of $\beta$-$Mg_2Si$, rather

than an equilibrium or stable β-Mg₂Si phase, was also consistent with another original research article by Sousa et al., wherein the authors noted the fact that the cooling rates achieved during gas-atomization of Al 6061 will not intersect the stable β-Mg₂Si cooling curves computed and presented in a plot of a computationally derived graphical depiction of the respective continuous cooling transformation (CCT) curves in [34].

**Table 1.** The average composition, in atomic percent, was assessed for each of the phases associated with Figure 4.

| Element | "White Phase" | "Gray Phase" | "Dark Phase" |
|---|---|---|---|
| Mg | 2.19 at. % | 2.56 at. % | 27.18 at. % |
| Al | 79.4 at. % | 89.3 at. % | 61.13 at. % |
| Si | 8.42 at. % | 3.71 at. % | 11.69 at. % |
| Fe | 9.67 at. % | 3.54 at. % | – |
| Cr | 0.15 at. % | 0.04 at. % | – |
| Zn | 0.1 at. % | – | – |
| Mn | 0.08 at. % | 0.03 at. % | – |

In any case, the identity of the white and gray phases was difficult to pinpoint in exact terms because of their respective similarity in chemical compositions and the stoichiometric similarity of the phases predicted by computational thermodynamic analysis, as detailed in [49,57]. Moreover, the lack of copper-EDS data associated with Figure 4 and Table 1 limited the ability to confidently assess copper's presence within two of the non-(β-Mg₂Si) intermetallic phases. Therefore, further TEM analysis was applied to another gas-atomized Al 6061 CSAM feedstock powder particle, approximately 47 μm in diameter, as shown in Figure 5. Again, equiaxed dendritic grains were found within the respective powder particle, with discrete smaller phases along the grain boundaries.

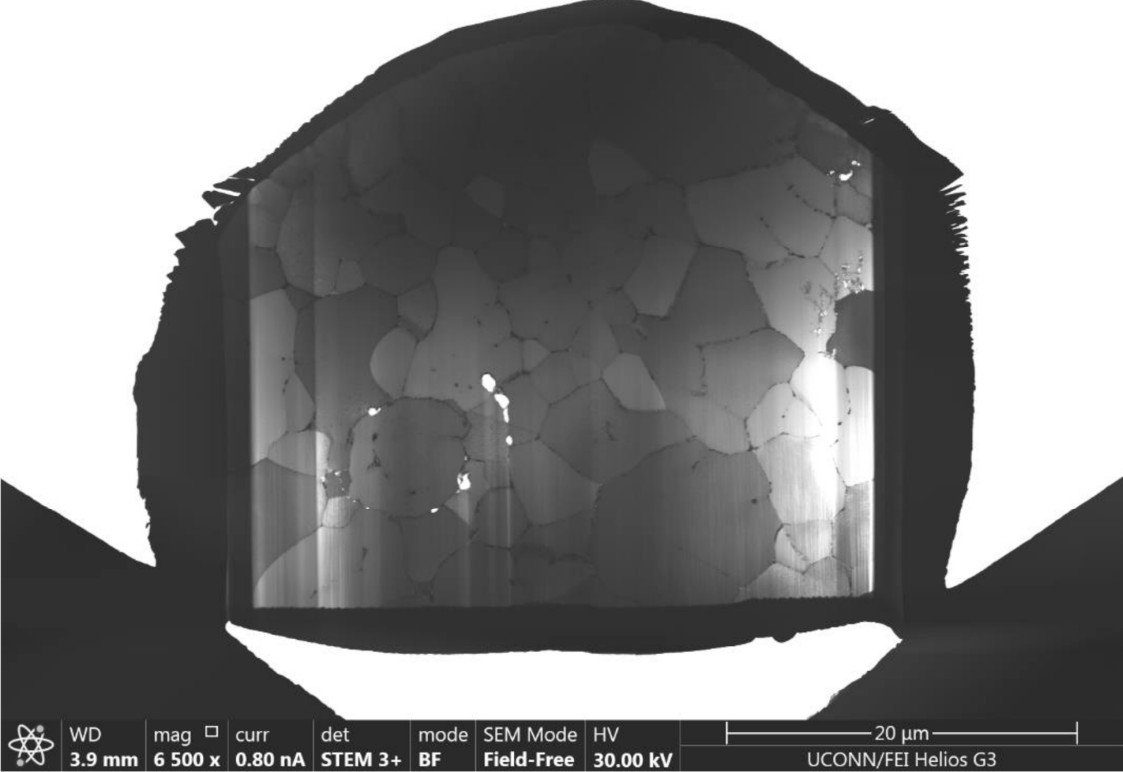

**Figure 5.** STEM micrograph of an as-atomized Al 6061 powder particle (with an approximately 47 μm diameter), prepared for STEM-based characterization via FIB milling.

The micrograph presented in Figure 5 highlighted how an equiaxed or compound dendrite-like microstructure within gas-atomized Al 6061 feedstock could be observed. Said solidification structure observed within the particulate captured in Figure 5 was consistent with the work of Molnárová et al. for another gas-atomized aluminum alloy [58]. As alluded to, the micrograph presented in Figure 5 also captured the location of secondary intermetallic phases along the grain boundaries in the gas-atomized Al 6061 powder. However, the need for a more sensitively quantitative analysis of regional and site-specific compositions remained. To address such a need, EDS mapping of various regions of interest within the gas-atomized Al 6061 FIB lamella shown in Figure 5 was performed. The EDS maps of said regions of relevance are shown in Figures 6–8. Once again, one ought to note that copper was not included in the EDS maps because of the influence that would follow from the copper grid upon which the lamellae was seated. Figure 6 showed interesting precipitate structures along grain boundaries. An Mg-Si precipitate appeared to be a "backbone" type structure, which was identified as a Chinese script eutectic non-stable form of β-Mg$_2$Si, with primarily Al-Fe intermetallic particles within the spaces between the "spines" of the non-stable form of a β-Mg-Si precipitate.

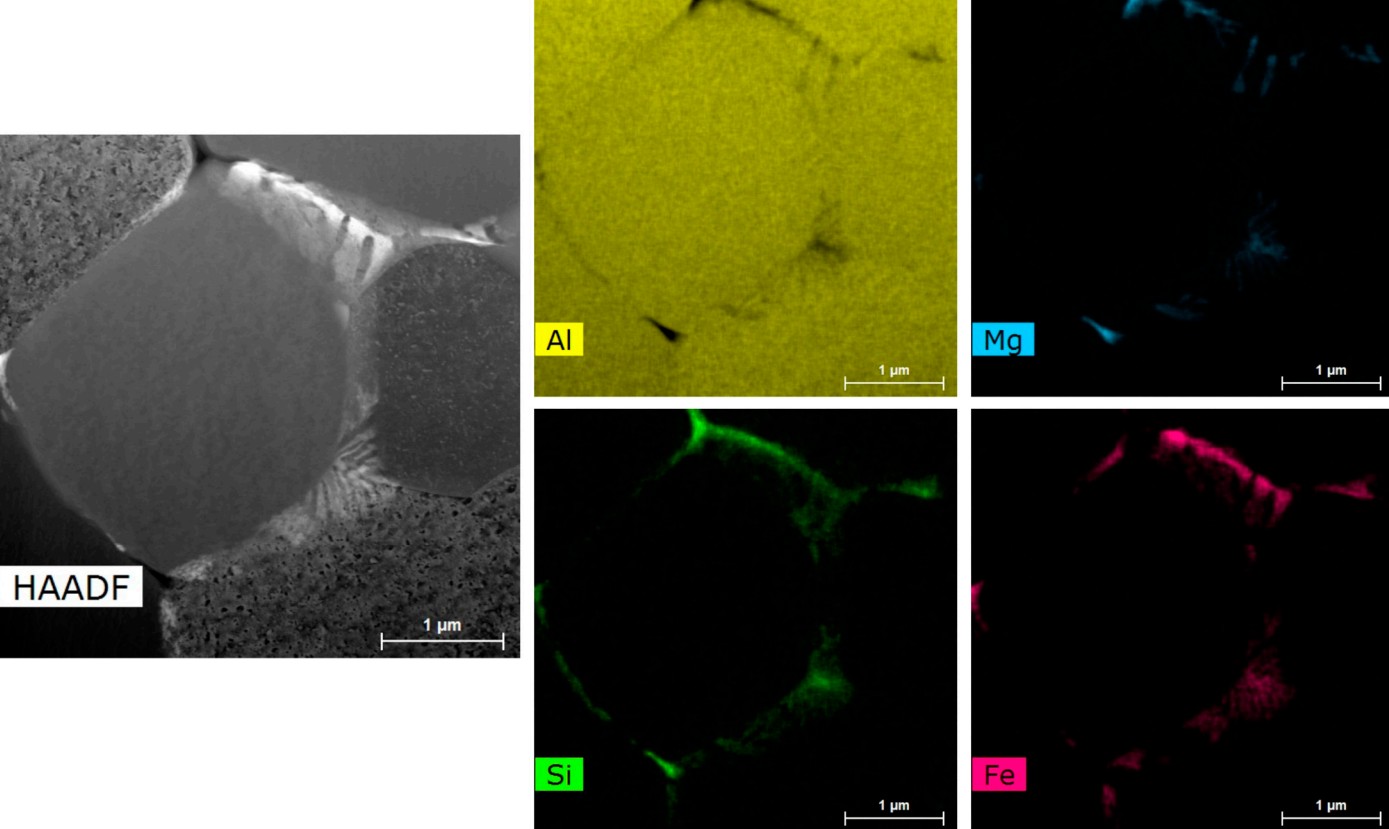

**Figure 6.** The subfigure labeled with "HAADF" presents a HAADF-STEM micrograph affiliated with a subset of subgrains and sub-granular boundary regions for an as-atomized Al 6061 powder particle. The colored micrographs labeled with Al, Mg, Si, and Fe present the respective EDS-derived elemental maps for the area presented in the HAADF micrograph. Note that the scale bars are all 1 µm.

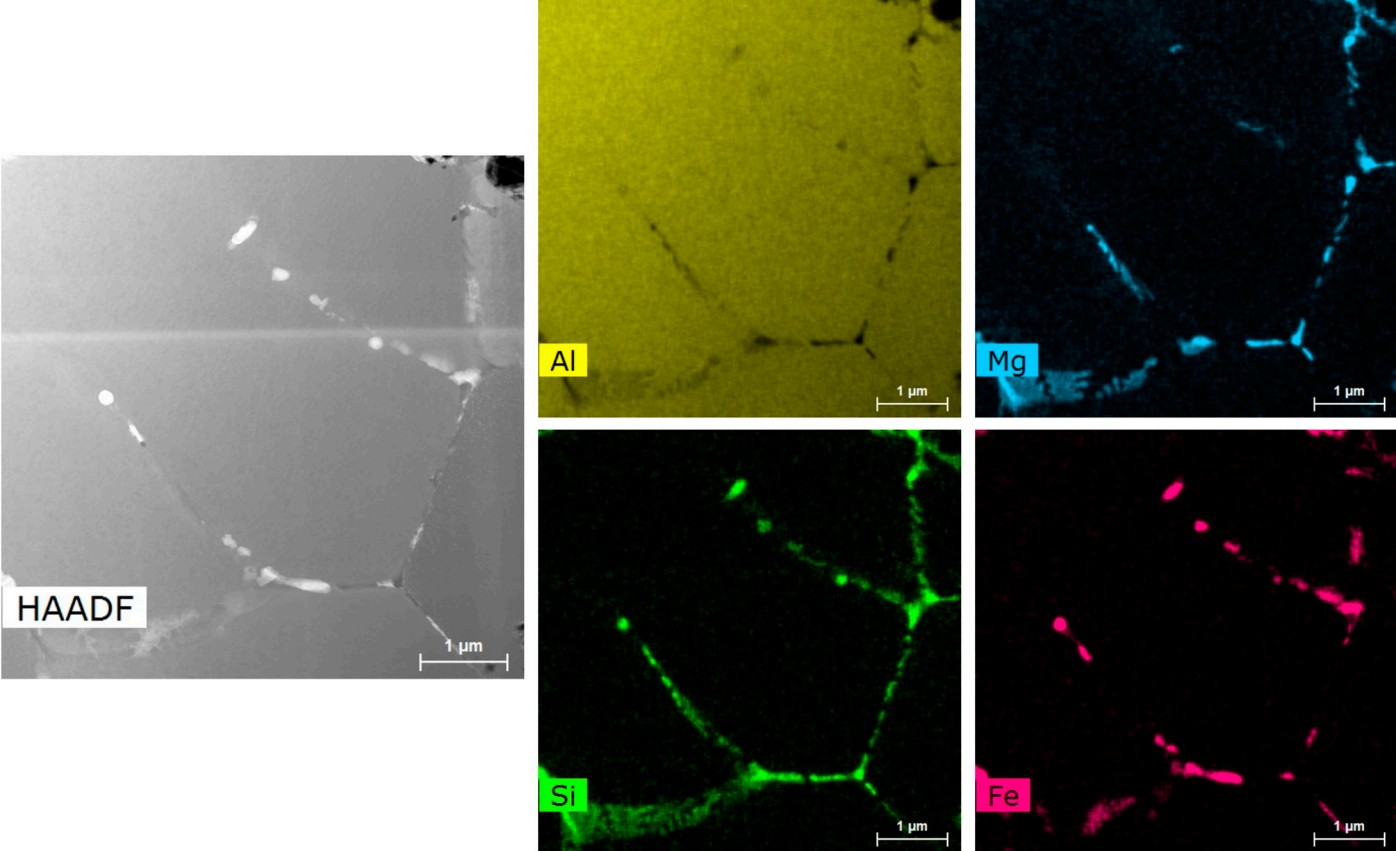

**Figure 7.** The subfigure labeled with "HAADF" presents a HAADF-STEM micrograph affiliated with a collection of subgrains and sub-granular boundary regions (emphasizing the discrete particles along the boundaries) for an as-atomized Al 6061 powder particle. The colored micrographs labeled with Al, Mg, Si, and Fe present the respective EDS-derived elemental maps for the area presented in the HAADF micrograph. Note that the scale bars are all 1 μm.

An inclined grain boundary was observed and recorded in Figure 8 and contains the same interesting "backbone" structured Mg-Si precipitates noted in Figure 6, i.e., the Chinese script eutectic non-stable form of β-Mg$_2$Si. The Mg-Si precipitate structure at said grain boundaries was dendritic, with the Al-Fe particles located in between the secondary dendrite arms. This was not found in the first STEM sample shown in Figure 4; however, it was evident across the sample introduced initially in Figure 5. Notably, the recorded microstructural intermetallic arrangements and morphologies were like the structure of the precipitates presented by Chakrabarti et al. in [16]. In addition, though there were still similarly dendritic Mg-Si precipitates captured within Figure 7, there were also observable and discrete Al-Fe particles that did not reside between the secondary dendrite arms but were still contained along the polycrystalline grain boundaries.

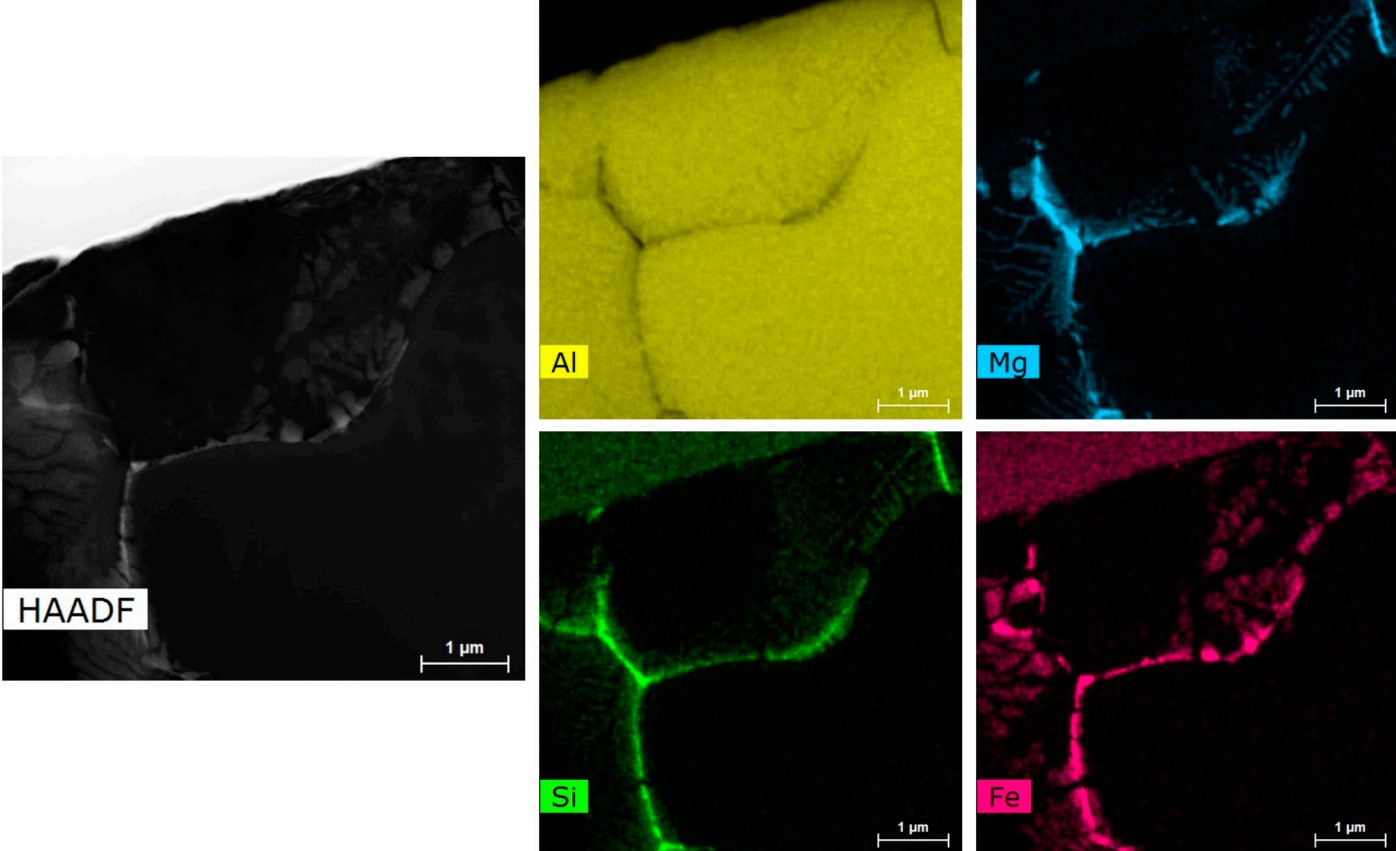

**Figure 8.** The subfigure labeled "HAADF" presents a HAADF-STEM micrograph affiliated with two larger polycrystalline boundary regions for an as-atomized Al 6061 powder particle. The colored micrographs labeled with Al, Mg, Si, and Fe present the respective EDS-derived elemental maps for the area presented in the HAADF micrograph. Note that the scale bars are all 1 μm.

Image analysis was performed upon the EDS mapping areas, i.e., locations captured in Figures 6–8, and their respective micrographs taken at several locations within the TEM sample presented in Figure 5. A histogram of the intermetallic size results of the Mg-Si and Al-Fe particles is presented in Figure 9. It was shown that most Mg-Si particles were around 40 nm to 80 nm in size. The Mg-Si phase was also found to yield an approximately 2.7% +/− 0.7% area fraction from the images analyzed. The Al-Fe intermetallic phases seemed to range widely in size, from 20 nm to over 140 nm, and were also found to yield an approximately 2.25% +/− 0.01% area fraction. Following the observable solidification structures documented for gas-atomized aluminum alloys, the interesting Chinese script-like eutectic structure of the non-stable form of β-$Mg_2Si$ precipitates ought to be considered dendritic; consequently, a secondary dendrite arm width and spacing were able to be reported within the regions of relevance as well. The secondary dendrite arm widths were measured and found to be approximately 53 nm +/− 16 nm, while the dendritic SDAS within said regions was measured as 78 nm +/− 17 nm. Such sub-100 nm sized SDAS and SDA values suggest that nanostructured regions exist within the gas-atomized Al 6061 feedstock particulates.

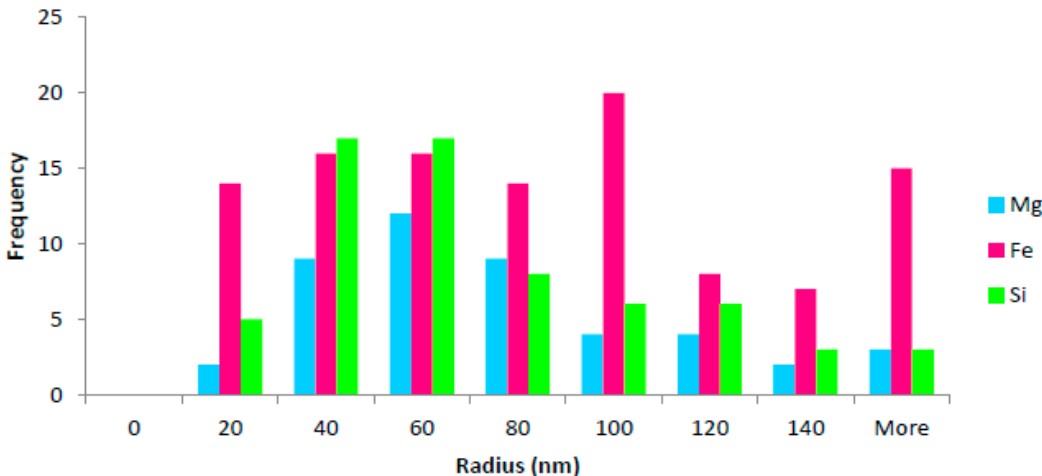

**Figure 9.** Image analysis was performed on the EDS mapping areas taken at several locations in the TEM sample. A histogram of the size results of the Mg-Si and Fe particles is shown herein. Accordingly, the precipitate radius size distribution of Mg, Si, and Fe intermetallic phases within an as-atomized Al 6061 powder particle is presented.

The solutionized sample was scanned in the DSC, described in the Methods and Materials section previously documented, and held at 540 °C for 1 h. The DSC curve is plotted in Figure 10. The typical peaks for the precipitation and dissolution of the β phase were found, like the literature data in Figure 10. A powder particle approximately 40 μm in diameter was examined in STEM and shown in Figure 11. Fairly equiaxed grains are found in the sample, with precipitates speckled at the grain boundaries. Upon closer inspection of the right side of Figure 11, there are bright, small precipitates containing heavier elements and larger dark precipitates at triple points of the grain boundaries. EDS was used to probe the identities of these phases once again. The DSC curve presented in Figure 10 was consistent with the work of Sousa et al. in [12].

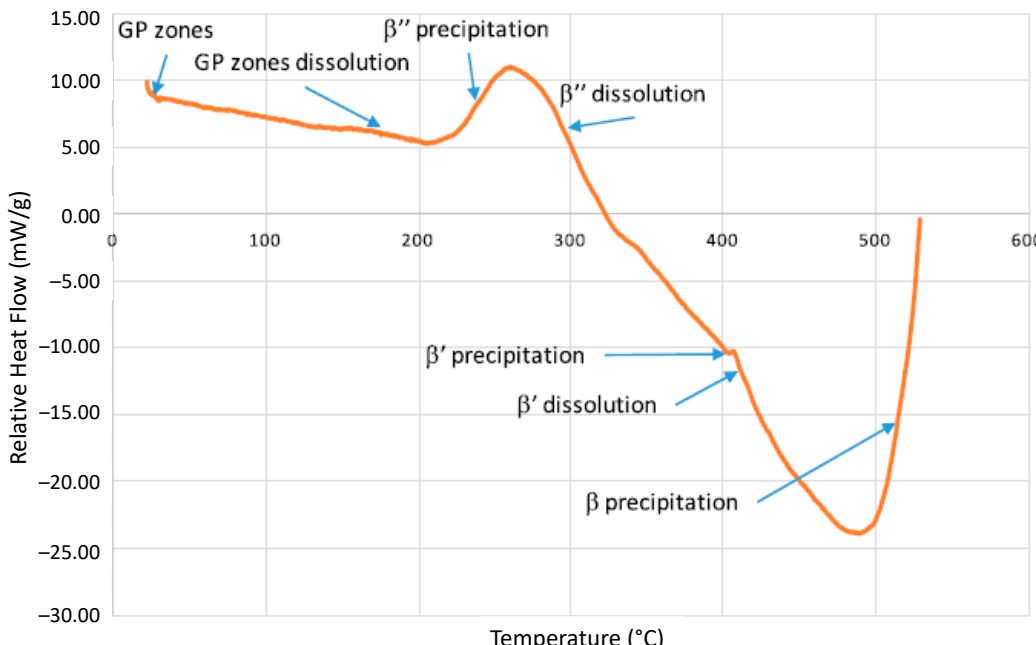

**Figure 10.** DSC scan for Al 6061.

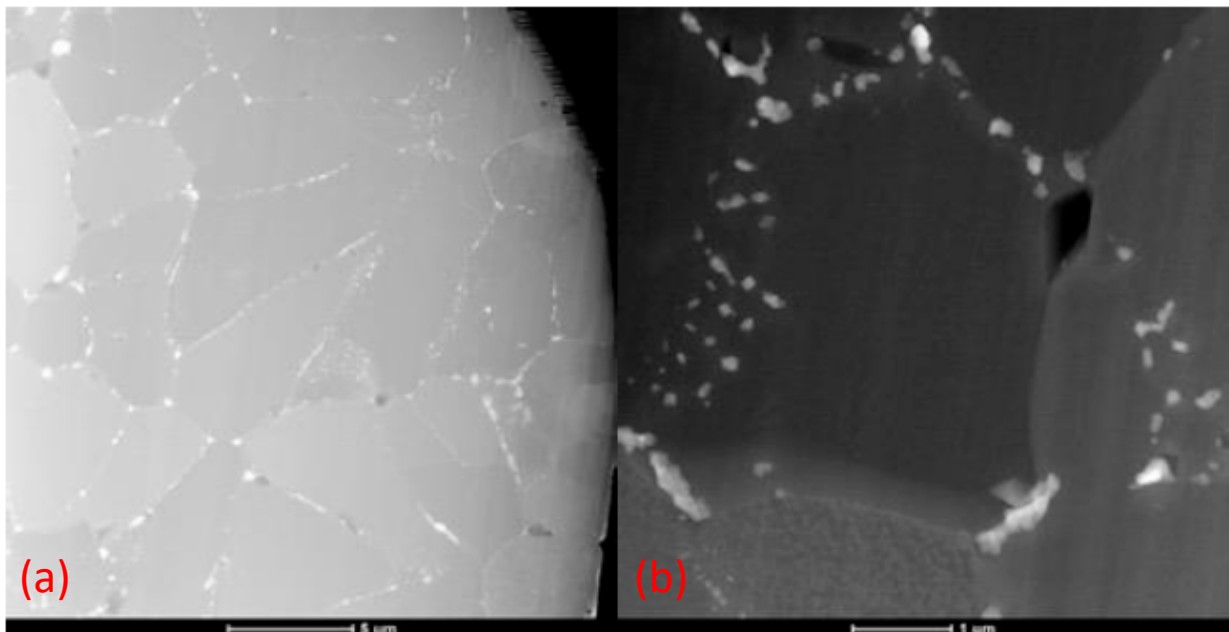

**Figure 11.** (**a**) A HAADF-STEM micrograph of a thermally pre-processed and gas-atomized Al 6061 powder particle was heat-treated at 540 °C for 60 min and prepared by FIB milling and polishing. (**b**) A higher magnification HAADF-STEM micrograph of the heat-treated Al 6061 gas-atomized powder particle wherein precipitates and intermetallic phases along the sub-grain boundaries can be observed. Note that the scale bar associated with (**a**) is 5 μm while the scale bar associated with (**b**) is 1 μm.

Through EDS mapping in Figures 12 and 13, it was found that the small bright white precipitates at the grain boundary contain almost all of the elements in the alloy; aluminum, silicon, iron, chromium, and manganese. This means these phases are difficult to identify and could be any combination of the following phases from the predictive tools: $Al_{13}Fe_4$, $Al_8Fe_2Si$, $Al_{15}Si_2Fe_4$, $Al_9Fe_2Si_2$, and $Al_{18}Fe_2Mg_7Si_{20}$. The darker, larger precipitates mostly contained magnesium and silicon at triple points and were hypothesized to be $Mg_2Si$. The precipitate radius size distribution of Al-Fe-Si-Cr-Mn and $Mg_2Si$ particles is shown in Figure 14. It is found that the Al-Fe-Si-Cr-Mn precipitates are small, with the majority around 40 nm to 60 nm and with a few over 200 nm in size. The average area fraction of these particles is approximately 4.1 +/− 1.3%. The $Mg_2Si$ phase has seemed to be lower in number but coarser in size. Most of the precipitates are over 200 nm, with an average of 324 +/− 154 nm in size.

There was a significant change in the morphology of the phases during the solutionization step. Firstly, the Al-Fe-Si-Cr-Mn particles seemed to have refined in size. There was an extensive size range in the as-received sample, from 20 nm to over 140 nm in size. In the solutionized sample, most of these particles were around 40 nm to 60 nm, with only a few over 200 nm in size. Secondly, the solutionization step dissolved the interesting dendritic structure of the $Mg_2Si$ phase. The non-dendritic precipitates found in the as-atomized sample were around 40 nm to 80 nm in size, with a 2.7 +/− 0.7% area fraction. After solutionization, the $Mg_2Si$ phase does not contain any precipitates in the dendritic structure, but it lessened in number and became coarser in size. Most of these precipitates were between 200 nm and 500 nm in radius. The sample had approximately 15 $Mg_2Si$ precipitates, mostly at triple points.

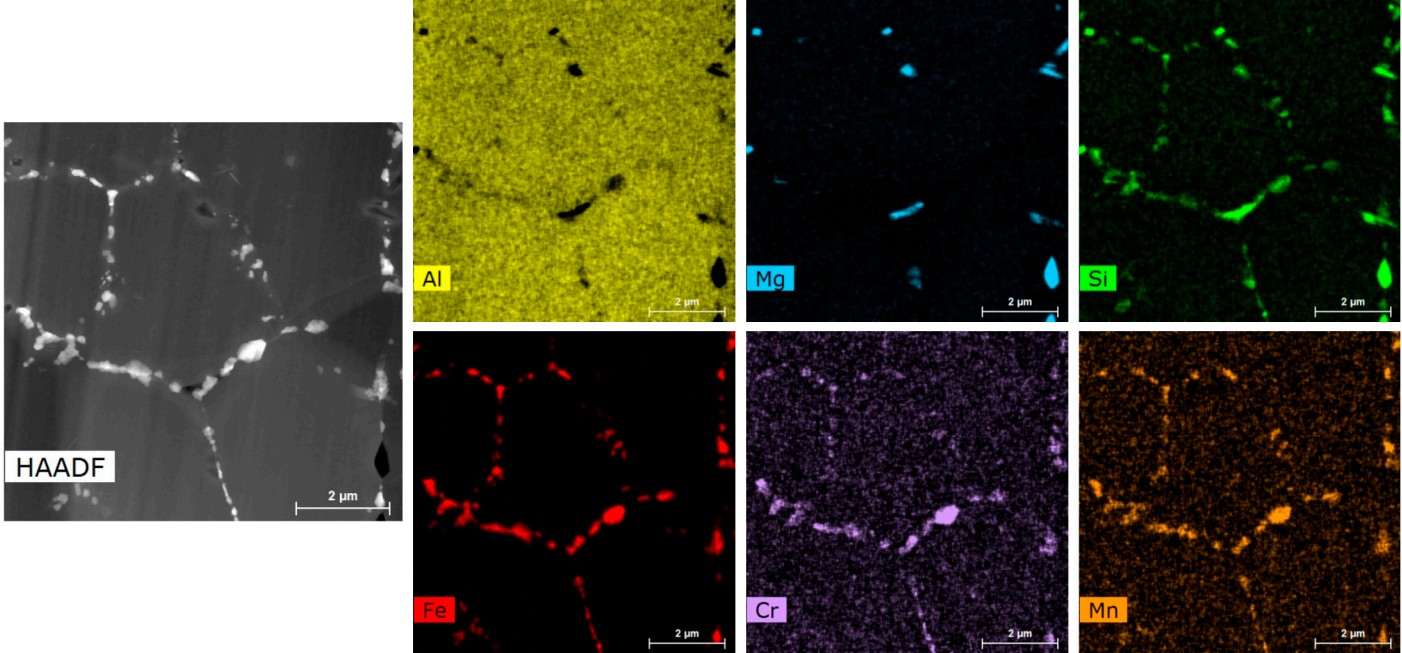

**Figure 12.** The subfigure labeled with "HAADF" presents a HAADF-STEM micrograph affiliated with a collection of subgrains and sub-granular boundary regions, with particular attention afforded to the discrete intermetallic precipitates along the polycrystalline boundaries within a thermally pre-processed Al 6061 gas-atomized powder particle. Thermal pre-processing was achieved via a heat treatment hold time of 60 min and a hold temperature of 540 °C. The colored micrographs labeled with Al, Mg, Si, Cr, Mn, and Fe present the respective EDS-derived elemental maps for the area presented in the HAADF micrograph. Note that the scale bar affiliated with each micrograph is 2 μm.

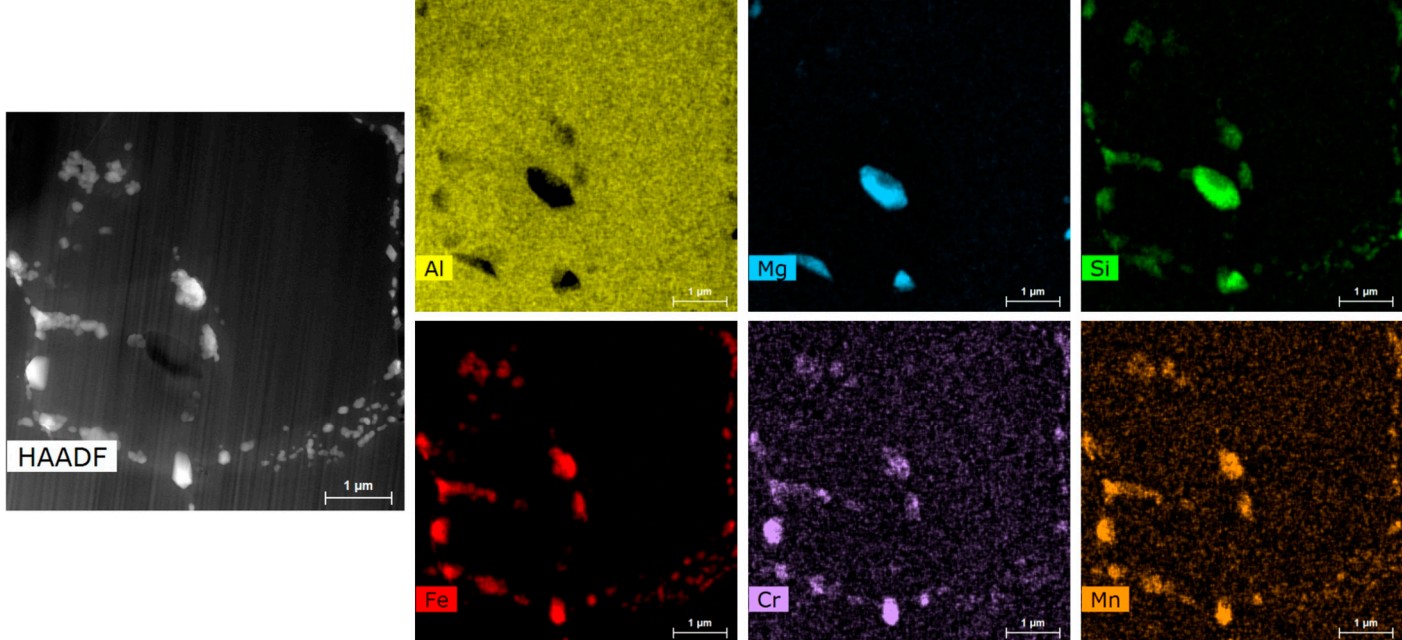

**Figure 13.** HAADF STEM image of discrete Fe-containing particles and larger Mg-Si particles at grain boundaries of heat-treated Al 6061 (540 °C for 1 h); EDS maps of Al, Mg, Si, Fe, Cr, and Mn.

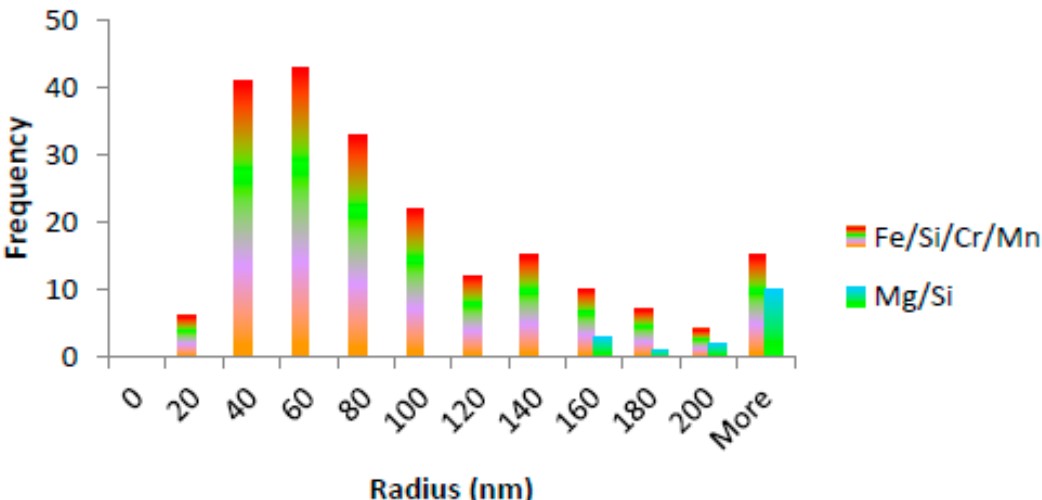

**Figure 14.** Precipitate radius size distribution of Al-Fe-Si-Cr-Mn and Mg-Si particles for heat-treated Al 6061 (540 °C for 1 h).

Nanoindentation was applied to the gas-atomized Al 6061 powder, both with and without thermal processing. The average nanoindentation hardness ($H = P/A_c$, wherein $A_c$ is the contact area function dependent upon the reporting indentation depth) of the as-atomized Al 6061 powder, with a size range of 38–45 µm was 1.12 GPa, and an average nanoindentation hardness value of $1.26 \pm 0.15$ GPa for randomly selected particulates of variable size categories, i.e., within the un-sieved powder. The variation in nanoindentation hardness values for difference solutionization hold times at an elevated temperature of thermal processing temperature of 540 °C was recorded as follows: 1.29 GPa at zero minutes, 1.23 GPa at 20 min, 1.19 GPa at 60 min, 1.14 GPa at 120 min, and 0.88 GPa at 240 min. It was shown that an initial increase in nanoindentation hardness existed at 0 h when the sample was brought to 540 °C and immediately quenched. As the time held at the solutionization temperature increased, the nanoindentation hardness tended to decrease, with the lowest value after 4 h of elevated temperature exposure. This could have been due to the coarsening and loss of coherency with the main strengthening phase, $Mg_2Si$. These results show that heat treating the powder will reduce its hardness, thus reducing the flow stress of the material and making it more "sprayable".

Power law curve fitting was applied to a multi-curve fit dataset obtained using each loading cycle load–depth data captured via nanoindentation testing as a function of specimen solutionization condition. Accordingly, the load–depth data were found to take the following form: $P = C(h_c)^m$, wherein the magnitude of the coefficients (*C*) were found to be 0.00003, 0.0005, 0.00006, 0.0002, and 0.000002 for the 0, 20, 60, 120, and 240 min solutionized powder particles, respectively. In said order, the power exponents (*m*) were determined to be 1.9912, 1.4508, 1.8486, 1.6753, and 2.4268, respectively. Finally, the statistical regression scoring metric known as the coefficient of determination ($R^2$) was also computed per hold time, in the same order, yielding 0.9959, 0.9987, 0.9970, 0.9979, and 0.9937, respectively. Finally, Figure 15 presents the power-law curve fit load–depth data associated with the loading portion of the test for each solutionization hold time to provide additional contextualization of the nanoindentation-based mechanical behavior across each condition studied.

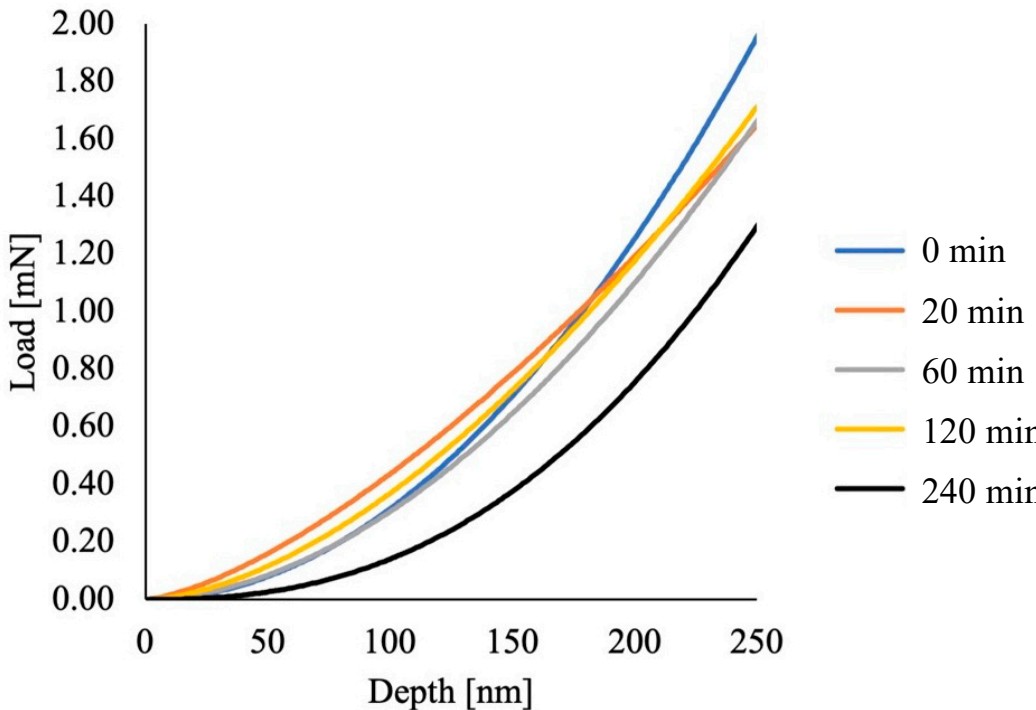

**Figure 15.** Average power-law fit load–depth curves for the studied and thermally pre-processed (and as-atomized) conditions.

Since the point in time when cold spray processing was first reported within the early scholastic literature of relevance, let alone cold spray's recent and formalized extension into the realm of metal additive manufacturing via CSAM, numerable mechanistic models and relations have emerged relating impact phenomena with material properties and the like. For example, CSAM researchers not only modified [59] and applied [60] the Johnson–Cook plasticity model for computationally exploring impact-induced mechanical, microstructural, and thermal evolution, as highlighted in [61] through [62]. Instead, the materials research community focused upon CSAM processing development also invoked, recalled, and refined numerable relations between the following, for instance: the critical impact velocity and the ultimate tensile strength of a microparticle [63]; the critical impact velocity and the yield/flow stress of a microparticle [64]; the strain rate experienced during particle impact and the microhardness of the material [33]; the dependence of deposition efficiency on particle size and thermophysical properties of the powder; the critical impact velocity as a function of spall strength, the bulk modulus and speed of sound, etc. [65]; the critical impact velocity as a function of oxygen content in gas-atomized pure Cu, for example [36]; and the mean impact pressure as a function of particle impact velocity; among others.

Of course, such an itemization only captures a few of the notable developments made to date following the continued evolution of our collective understanding of the feedstock material properties and microstructural relations underpinning and CSAM processing and thus resultant consolidated component performance. In so far as such mechanistic developments underlying CSAM are related to the experimental results presented and discussed herein, such formulations and theoretical, computational, or empirical relations can be coupled with said findings better to understand their implications for CSAM processing optimization and tunability.

## 4. Conclusions

Previous success in CSAM processing of powder in a solutionized condition has led to more research in thermally pre-processing powder before being processed via CSAM. An optimal heat treatment schedule is still needed, as it will differ from the traditional heat

treatment schedules reported in the literature for bulk Al 6061 counterparts. Considering the potential suitability of a homogenized microstructure CSAM, future work surrounding the continued development of dissolution models will aid in predicting when the soluble solute atoms have gone into solution and the dissolvable precipitates have been effectively eliminated. Such a model could be analytical or utilize commercially available diffusion software such as DICTRA (Thermo-Calc Software, Stockholm, Sweden). More experiments on heat treatment time and temperature need to be conducted. This work provided another preliminary exploration of the microstructural effects when exposed to high temperatures for a short period for gas-atomized Al 6061 CSAM feedstock.

Specifically,

- Consistent with the general understanding of impact-induced cold spray-related phenomena, XRD characterization and analysis revealed no measurable deformation- or processing-induced secondary phase transformations pre- and post-deposition of the as-atomized and cold sprayed Al 6061 specimens.

- Given the degree of variability in the interpretation of microstructural classification surrounding rapidly solidified and gas-atomized Al alloys, EBSD analysis was applied to the as-atomized Al 6061 powder. EBSD analysis demonstrated the presence of both high- and low-angled grains, wherein high-angled grains were comprised of similarly orientated sub-grains, which was found to be consistent with either compound/equiaxed/dendritic solidification.

- STEM analysis unveiled at least three secondary intermetallic precipitates, which generally ranged from 100–200 nm in size and were primarily observed along the polycrystalline microstructural grain boundaries. In the as-atomized powder, the three phases observed (in addition to the Al matrix phase) included a $\beta$-$Mg_2Si$-type intermetallic; Al-Fe-type intermetallics; and Mg-Si-type intermetallics. Identification was achieved via HAADF-STEM and STEM-EDS analysis.

- DSC analysis of the as-atomized powder unveiled the following dissolution/precipitation sequence: GP zone dissolution $\rightarrow$ $\beta''$ precipitation $\rightarrow$ $\beta''$ dissolution $\rightarrow$ $\beta'$ precipitation $\rightarrow$ $\beta'$ dissolution $\rightarrow$ $\beta$ precipitation.

- HAADF-STEM and EDS analysis of thermally pre-processed Al 6061 powder particles demonstrated the presence of two dominant categories of intermetallic phases present after applying the 540 °C for 60 min heat treatment: Al-Fe-Si-Cr-Mn-type and Mg-Si-type constituents. Moreover, the Al-Fe-Si-Cr-Mn-type phases were predominately smaller than those maintained by the Mg-Si-type intermetallic phases.

- By way of applying suitable nanoindentation testing protocols, the measured hardness at a constant reporting depth was found to decrease as a function of thermal pre-processing hold time, ultimately dropping from a mean 250 nm depth-based hardness of 1.29 GPa after ramping up to 540 °C and followed by immediate quenching (i.e., a hold time of 0 min) to 0.88 GPa after 240 min. In turn, insights into pre-processing structure–hardness relations were able to be gleaned.

- Concerning cold spray additive manufacturing, work by Lima et al. suggests that the degree of impacted particle flattening (or straining) introduced is inversely proportional to the particulate yield stress, or hardness of the feedstock, under constant cold spray processing parameters [66]. Therefore, microstructure-to-plasticity linkages will be explored in future work wherein pre-processing coupled with constant cold spray processing parameters will be consolidated and characterized to enhance the through-process understanding of solid-state additive manufacturing of Al 6061 via cold spray.

As far as the future work surrounding the development of a suitable dissolution model, experiments explicitly designed for comparison with the prospective kinetic models will be essential. Characterization techniques that have proved successful for these experiments were S/TEM with image analysis of solutionized samples to determine phase fractions in comparison with the as-atomized condition and nanoindentation-based characterization for mechanical property (hardness) assessment. Another interesting technique that ought

to be pursued in future work concerned with thermally processing CSAM feedstock is that of atom probe tomography. Atom probe tomography would be even more beneficial than EDS in STEM alone since EDS-STEM has revealed what elements are in each precipitate; however, it was challenging to identify the exact precipitates when they contained several alloying elements that are each able to be present in different intermetallic phases. Atom probe tomography will provide more information surrounding the composition of the micrometric and nano-sized precipitates and the coherency of said precipitates with the matrix phase. The much-needed and future thermodynamic and kinetic models, when utilized in conjunction with STEM and phase identification, will also enable more accurate and telling single-particle and multi-particle CSAM impact models to be formulated and probed such that one may more directly study and inspect the effects of precipitates along the interior grain boundaries of feedstock particulates during deposition. In short, CSAM particle impact models will likely become more accurate when microstructural features are implemented, as was briefly demonstrated in [12,36].

One of the primary goals of the present work was to characterize gas-atomized aluminum 6061 powder fully and quantitatively inspect the resultant microstructural evolution incurred through the lens of CSAM suitability. The basic microstructural research of aluminum particles will significantly contribute to powder metallurgy and additive manufacturing. Powder microstructures of aluminum differ significantly from their wrought and cast counterparts because of the unique atmosphere and solidification conditions during atomization. There is very little to no literature data on the microstructures of these alloys in powder metallurgy, particularly those of the powders of interest to this research. Initial solutionization heat treatments were performed on both alloys, and the samples were fully characterized and compared to the as-atomized powder. For Al 6061, grain growth was not significantly identifiable after applying the solutionization heat treatment. The microstructure, however, changed significantly. For Al 6061, the Al-Fe-Si-Cr-Mn intermetallic phase or phases refined in size after solutionization. The Chinese script eutectic non-stable $Mg_2Si$ phase was dissolved, but a few larger $Mg_2Si$ particles, between 200 and 500 nm in radius, formed and remained.

This work demonstrated the effects of heat treatments on aluminum powders. With more research, a complete understanding of the effect of thermal processing on Al 6061 and other alloyed powders will benefit the powder metallurgy community and the CSAM sector. The microstructure of the powders for additive manufacturing has yet to be the focus of research, as in most additive manufacturing techniques, particle melting usually occurs. Understanding the effects of the heat treatments will optimize CSAM and other solid-state processing and properties with heat-treated powders. Much success has already been accomplished using only trial-and-error methods at this point. The modeling and characterization techniques in the present work could easily be applied to different material systems.

**Author Contributions:** Conceptualization, B.H., B.C.S., K.T., V.K.C.J., R.D.S.J., A.N. and D.L.C.; data curation, B.H. and D.L.C.; formal analysis, B.H., B.C.S., R.D.S.J. and D.L.C.; funding acquisition, R.D.S.J. and D.L.C.; investigation, B.H., B.C.S., K.T. and D.L.C.; methodology, B.H., R.D.S.J. and D.L.C.; project administration, V.K.C.J., R.D.S.J., A.N. and D.L.C.; resources, V.K.C.J., R.D.S.J., A.N. and D.L.C.; software, B.H., K.T. and D.L.C.; supervision, V.K.C.J., R.D.S.J. and D.L.C.; validation, B.H., B.C.S. and K.T.; visualization, B.H.; writing—original draft, B.H., B.C.S. and K.T.; writing—review and editing, B.H., B.C.S., K.T., V.K.C.J., R.D.S.J., A.N. and D.L.C. All authors have read and agreed to the published version of the manuscript.

**Funding:** This work was funded by the United States Army Research Laboratory, grants #W911-NF-19-2-0108 and #W911-NF-10-2-0098.

**Data Availability Statement:** Not applicable.

**Acknowledgments:** B.H. wishes to thank Jianyu Liang, Satya Shivkumar, and Timothy Eden for their support and guidance. B.H. also wishes to recognize colleagues who offered constructive feedback and suggestions, including Aaron Birt, Luke Bassett, Rachel White, Caitlin Walde, Derek Tsaknopolous, Rob DelSignore, and Matt Gleason.

**Conflicts of Interest:** The authors declare no conflict of interest. The funders had no role in the design of the study, in the collection, analyses, or interpretation of data, in the writing of the manuscript, or in the decision to publish the results.

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
