# Peer review of "Thermal Preprocessing of Rapidly Solidified Al 6061 Feedstock for Tunable Cold Spray Additive Manufacturing"

_metals, doi:10.3390/met12071214_

Round 1

Reviewer 1 Report

This paper investigated the Al powders for a cold spray additive manufacturing , the as-atomized  condition and the thermally treated condition. The properties and microstructures of the powders were explored using various techniques, including STEM, XRD, SEM, EBSD, EDS, and DSC, and it was shown that the presence of Mg-Si and Al-Fe phases within the powder microstructures.

These results and discussion are very useful for the development of the CSAM, therefore this paper should be accepted for this journal.

(1)The results of nanoindentation measurements of powders are very important, so please show them graphically, not just in text.

(2)In the conclusion section, please add a more detailed statement summarizing the experimental results and discussion of this paper.

Reviewer 2 Report

This work is interesting and important for cold spraying aditive manufacturing, but there are several drawback should be calrified.

1 The line 40 claimed that particle with diameter less than 20 μm is unable to reach the critical velocity,. This illustration is a bit different from what I had learnt about cold spray, which is also different from the literatures cited here.

2 Although controlling the feedsock powder is important for controlling the cold spray depositing part, but this paper make a conclusion that is not supported by the experiment result. the author should deposit the treated powders to veriy their judgement instead of from cited literatures.

3the abstract and the should obey rule of the journal. it is more like a experiment and background introduction.
